# Integrated multi-omics approach revealed *TTNtv* c.13254T>G causing dilated cardiomyopathy in mice

Dan Yu[1], Liang Tao[1], Laichun Song[1], Kaisheng Lai[2], Hui Jiang[2], Zhe Liu[2], Hongyan Xiao [1]*

1 Division of Cardiac Surgery & Wuhan Clinical Research Center for Cardiomyopathy, Wuhan Asia Heart Hospital Affiliated with Wuhan University of Science and Technology, Wuhan, Hubei, P.R. China,
2 Department of Science Research Centre, BestNovo (Beijing) Medical Laboratory, Beijing, P.R. China

* xiaohy11@sina.com

**Data Availability Statement:** The sequencing data has been submitted to DRYAD database with DOI https://doi.org/10.5061/dryad.tmpg4f563 and can be accessed with the URL https://datadryad.org/

## Abstract

Titin-truncating variant (*TTNtv*) is the most common genetic cause of dilated cardiomyopathy (DCM). In the previous study, we found a novel heterozygous *TTN*tv c.13254T>G (p. Tyr4418Ter) associated with DCM, but lacking functional evidence. The purpose of this study is to demonstrate the pathogenicity of *TTNtv* c.13254T>G. We constructed a mouse model with *TTN*tv Y4370* on exon 45 by CRISPR/Cas9-mediated genome engineering to imitate the *TTN*tv. c.13254T>G. Transmission electron microscope (TEM), immunohistochemistry, western blot (WB), Transcriptome sequencing (RNA-seq), and tandem Mass Tag (TMT) proteome analysis were performed on the mutant (KO) and WT mice cardiac tissue. Multi-omics association analysis was performed to observe the damages of cardiac tissue, and changes of inflammatory factors and Titin protein. TEM results showed that *TTNtv* Y4370* may lead to broken myofibrils, sparse myofilament structure, and broken Z-line and H-zone in many places of cardiac tissue of KO mice. Immunohistochemistry showed a significant increase in cTnT and TNF-α expression level in KO mice cardiac tissue. RNA-seq and TMT proteome enrichment analysis further strengthened that *TTNtv* Y4370* led to cardiac injury and inflammatory response in KO mice. In summary, *TTNtv* c.13254T>G contributed to the cardiac injury, inflammatory response and construct alterations in mice, that is *TTNtv* c.13254T>G may cause DCM in mice. These functional evidence of *TTNtv* c.13254T>G have important significance for follow-up genetic research of DCM in human.

## Introduction

Dilated cardiomyopathy (DCM) is a heterogeneous myocardial disease characterized by ventricular enlargement and cardiac dysfunction [1], with a prevalence rate of approximately 1/2500-1/250 [2, 3]. DCM is the main cause of heart failure (HF) and heart transplantation (HT). It is reported that about 2/3 patients with DCM died of pump failure, and about 1/3 patients died of sudden cardiac death. Although great progress has been made in the treatment of

**Funding:** This study is supported by Special Project of Hubei Provincial Health Commission (WJ2021M033). There was no other additional external funding received for this study. The author Dan Yu received the grant. Dan Yu participated in study design, conducted formal analysis and methodology. In addition, she wrote original draft and reviewed & edited the manuscript.

**Competing interests:** The authors declare no competing interests.

clinical drugs and assistive devices, the 5-year mortality rate of DCM patients still reaches 20% [4]. At present, 35% of DCM cases have been confirmed as familial inheritance. In DCM families, 26 chromosomal loci associated with the disease have been mapped by candidate gene screening and retrieval analysis, and more than 60 pathogenic genes have been identified [5].

*TTN* gene is the most common pathogenic gene of DCM [6], which consists of 363 exons (ENST00000589042), a variety of mRNA subtypes, and encodes the largest protein Titin in human body with 34,350 amino acids. Titin protein spans half of the sarcomere from the Z disc to the M line, and is related to sarcomere elasticity, assembly and mechanical induction [7–9]. Heterozygous titin-truncating mutation (TTNtv) is the most common genetic cause of familial DCM, accounting for about 25% of the cases [10]. Due to its huge size, *TTN* gene mutations, including truncation mutations and missense mutations, are relatively common in the general population.

In previous study, we reported two probands diagnosed with DCM carrying a novel *TTN*tv c.13254T>G (p.Tyr4418Ter) on exon 46 of TTN (the transcript was NM_133379.5) and constructed a mouse model with *TTN*tv Y4370* on exon 45 by CRISPR/Cas9-mediated genome engineering to imitate the newly discovered human *TTN*tv. Through echocardiography, serological detection and histological evaluation performed on KO and WT mice, we preliminarily indicated that *TTN*tv Y4370* mouse have typical cardiac changes of DCM [11]. To further supplement the pathogenic evidence of *TTN*tv c.13254T>G, transmission electron microscope (TEM), immunohistochemistry, western blot (WB), conjoint analysis of RNA-seq and TMT proteome were performed after euthanasia of mice and harvesting the heart in current study.

## Materials and methods

### Ethics statement

The study was conducted in accordance with The Code of Ethics of the World Medical Association. The mouse studies were performed in accordance with ethical guidelines of Wuhan Asia Heart Hospital, China. The experimental protocols were in accordance with the ARRIVE guidelines and were approved by the Wuhan Asia Heart Hospital Animal Care and Use Committee (Ethics reference number: 2021-YXKY-P007). All animal protocols conformed to the National Research Council's Guide for the Care and Use of Laboratory Animals. A total of 4 KO mice with heterozygous *TTN*tv Y4370* were obtained, and 4 even-aged WT mice were selected as control.

### Generation of mouse model

Sequence(GATGATGTTGGATATCATGGACCGGACTGGGGAAATATGAAAGGACATTCTCAAA GTGAT(TAT)GTGCTAAAT(ATC)CACTC//CAAGAGAACCTCTAACACAGTTCAGGACGTG GAAGACTCACCTGTCCCTACCTAT, "//" indicates gRNA cut site) on exon 45 of the mouse *TTN* gene was selected as the target site. gRNA-A1 sequence TGTTAGAGGTTCTCTTGGAG-TGG, and donor oligo with targeting sequence flanked by 120 bp homologous sequences combined on both sides were designed. Off-target analysis for gRNA-A1 was performed. Cas9, gRNA and donor oligo were co-injected into fertilized eggs of mice with C57Bl/6J background for F0 mouse production. A positive mouse with *TTN* p.Y4370* heterozygous mutation was selected to backcross with the C57Bl/6J background mouse for the production of F1 generation followed by genotyping.

## Methods of sacrifice

Mice were euthanized by cervical dislocation after anesthesia. Mice were subjected to induced anesthesia with 5% isoflurane until completely stationary, the corneal reflex disappeared, and no pain response was observed when the tail was clamped. Then, the anesthetized mice were placed on a wire mesh. The experimenter pulled the tail of the mouse with one hand, pressed the neck of the mouse with tweezers, and pulled the mouse with hands to dislocate the cervical spine.

## Methods of anesthesia and/or analgesia

The mice were anesthetized with a mixture of isoflurane and oxygen. After connecting the pipeline, the anesthesia induction mode with an isoflurane concentration of 3% was set, and the diverter was opened to input the mixture of isoflurane and oxygen into the anesthesia induction box for a 1-minute induced anesthesia. After induced anesthesia, the concentration of isoflurane was adjusted to 1.5% ~ 2% to maintain anesthesia.

## Efforts to alleviate suffering

Subcutaneous injection of 0.05–0.1mg/kg buprenorphine was carried out to relieve pain when abnormal pain or urgent response occurred in mice, and intraperitoneal injection of diazepam 5 mg/kg was applied for sedation when mice were nervous.

## Transmission Electron Microscope (TEM)

The fresh mouse heart tissue block was harvested minimizing mechanical damage such as pulling, contusion and extrusion. The heart tissue block was then immediately put into the Petri dish containing fixative (G1102, Wuhan Servicebio Technology Co., Ltd., China) for the electron microscope (HT7800/HT7700, Hitachi, Japan) and cut into small tissue blocks no more than 1 mm$^3$ by a sharp blade within 1–3 minutes. The 1mm$^3$ tissue blocks were transferred into an EP tube with fresh TEM fixative for further fixation, which was fixed at 4°C for preservation and transportation. And then the tissues were washed using 0.1 M PB (pH 7.4) for 3 times, 15 min each. Tissues were post-fixed avoiding light with 1% OsO4 (Ted Pella Inc, USA) in 0.1 M PB (pH 7.4) for 2 h at room temperature. After OsO4 was removed, the tissues were rinsed in 0.1 M PB (pH 7.4) for 3 times, 15 min each. After dehydrated embedding the tissue, the cuprum grids were observed under TEM (Hitachi, Japan) and taken images.

## Immunohistochemistry

Immunohistochemistry was applied to detected the difference of inflammatory factors IL-6, IL-10, TNF-α, TGF-β, MMP2, MMP9 and cTNT of cardiac tissue between KO mice and WT mice to validate whether cardiac structure alteration happened in KO mice. The cardiac tissue was dehydrated by 75% alcohol (10009218, Sinopharm Chemical Reagent Co., Ltd., China) for 4h, 85% alcohol for 2h, 90% alcohol for 1.5h, 95% alcohol for 1h, anhydrous ethanol I for 0.5h, anhydrous ethanol II for 0.5h. After dehydration, tissue blocks were transparent. The transparent tissue blocks were immersed in 60°C paraffin I for 1h, 60°C paraffin II for 1h, 60°C paraffin III for 1h. The tissue blocks soaked by wax (69019361, Wuhan Servicebio Technology Co., Ltd., China) were embedded in the paraffin block. After the sections were dried with absorbent paper, HRP-labeled goat anti-rabbit secondary antibody (DD13, Talentbiomedical, China) was added dropwise and incubated at 37°C for 30 min. The sections were rinsed with PBS for 4 times, 3 mins each, and the PBS solution was shaken off. Each section was added with freshly prepared DAB color developing solution (G1212-200T, Wuhan Servicebio Technology Co.,

Ltd., China). Under the microscope (DS-Fi3, Nikon, Japan), the positive signal was brownish yellow or brown. The sections were rinsed with tap water to terminate the color development. The slices were re-stained by Mayer hematoxylin (H9627, Sigma, Germany) for 2min, and then washed with PBS water back to blue. After the slices were rinsed in water, the slices were placed in 70% alcohol, 80% alcohol, 90% alcohol, 95% alcohol-anhydrous ethanol I, anhydrous ethanol II, xylene (10023418, Sinopharm Chemical Reagent Co., Ltd., China) I, xylene II for dehydration and transparency, 5 mins each, and finally air-dried in a fume hood. The neutral gum (10004160, Sinopharm Chemical Reagent Co., Ltd., China) was dropped next to the tissue, and then covered with a cover glass. The sealed sections were lying flat in a fume hood to dry. Dried sections were observed under a microscope.

## Western blot

Titin was the target protein. The supernatant protein was taken to measure the protein concentration using BCA reagent kit (Merck, Germany). The protein concentration of all samples was adjusted to 1.5ug/ul using RIPA (20–188, Merck, Germany). 30ug of protein was mixed with a 1/4 volume 5× loading buffer (P0015, CWBIO, China) of the sample, and heated in a 100°C water bath for 10 minutes. Electrophoresis was performed with 4–15% SDS-PAGE Gel (P0012AC, CWBIO, China) under 100V voltage stabilization for 1 hour. Until the sample entered the separation gel, the voltage was adjusted to 130V voltage stabilization until the loading buffer dye reached the bottom of the gel. After electrophoresis was stopped, the plate was disassembled, and wet transfer of the membrane was performed under the conditions of 25mM Trisbase (0497, Amersco, USA), 192mM Glycine (0167, Amersco, USA), 20% methanol (10014108, Sinopharm Chemical Reagent Co., Ltd., China) transfer buffer, 4°C, 100V stabilized conversion for 60 minutes. The membrane was sealed with 5% DifcoTM Skim Milk (232100, BD, USA) at room temperature for 1 hour. After sealing, the membrane was washed 3 times with TBST (72015460, Sinopharm Chemical Reagent Co., Ltd., China). The first antibody (titin, ab284860, Abcam, UK) is diluted 1000 times with 3% bovine serum albumin (BSA) (C134730100, Sinopharm Chemical Reagent Co., Ltd., China) configured with TBST. The membrane was incubated with diluted first antibody in a horizontal shaking bed (WD-9405F, BEIJING LIUYI BIOTECHNOLOGY CO., LTD., China) at 4°C overnight. After the incubation of the first antibody was completed, the membrane was washed with TBST 3 times, 10 minutes each. The second antibody (A0208, Beyotime Biotechnology, Beijing, China) was diluted 1000 times with 5% skim milk sealing solution and the membrane was incubated with the diluted second antibody at room temperature for 60 minutes. PVDF membrane (Millipore, USA) was washed with TBST washing 3 times, 10 minutes each. Then the PVDF membrane was lifted to drain any excess washing solution from the membrane. The film flat was spread on a clean cling film and placed in the multifunctional microplate reader (EnSpire, PerkinElmer, USA). The prepared ECL (32106, Thermo Scientific, USA) reaction solution was subsequently added for a 2-minute reaction before exposure for photography.

## Transcriptome sequencing

Total RNA was extracted using PureLink™ RNA Micro Kit (12183018A, Invitrogen, USA). The integrity was detected by Ailent 2100 Bioanalyzer (Agilent Technologies Inc, USA) using RNA 6000 Nano kit 5067–1511 (Agilent Technologies Inc, USA). The mRNA with polyA structure in total RNA was enriched by Oligo (dT) magnetic beads, and the RNA was broken to a fragment of about 300 bp by ion interruption. Library was prepared by NEBNext Ultra II RNA Library Prep Kit for Illumina (E7770L, New England Biolabs Inc, USA). The first strand of cDNA was synthesized using RNA as template, 6-base random primers and reverse

transcriptase, and the second strand cDNA was synthesized using the first strand cDNA as template. After the library construction was completed, the library fragments were enriched by PCR amplification, and then the library was selected according to the fragment size. The library size was 450 bp. Then, the quality of the library was tested by Agilent 2100 Bioanalyzer (Agilent Technologies Inc, USA) using Agilent High Sensitivity DNA Kit (5067–4626, Agilent Technologies Inc, USA). The total concentration of the library and the effective concentration of the library were detected by Qubit® 2.0 Fluorometer (Invitrogen, USA) and Agilent 2100 Bioanalyzer. Then, according to the effective concentration of the library and the amount of data required for the library, the libraries containing different Index sequences (each sample was added with different Index, and finally the offline data of each sample is distinguished according to the Index) were mixed in proportion. The mixed library was uniformly diluted to 2nM, and the single-strand library was formed by alkali denaturation. The libraries were sequenced by Next-Generation Sequencing (NGS) based on Illumina NovaSeq 6000 platform (Shanghai Personal Biotechnology Cp. Ltd, China) in PE150 mode.

## Tandem Mass Tag (TMT) proteome

Appropriate amount of protein was added to a final concentration of 5mM DTT (M109-5G, Amresco, USA) incubated at 37°C for 1h, and then returned to room temperature. A final concentration of 10 mM iodoacetamide (M216-30G, Amresco, USA) was added and incubated at room temperature for 45 min in the dark. The sample was diluted 4 times with 25mM ammonium bicarbonate (A6141-500G, Sigma-Aldrich, Germany) and trypsin (V5280/100ug, Promega, USA) was added according to the ratio of protein to trypsin 50:1, and incubated overnight at 37°C. On the second day, formic acid (T79708, Sigma-Aldrich, Germany) was added to adjust the pH to less than 3, and the digestion was terminated. The sample was desalted using a C18 desalination column (186003581, Waters, USA), 100% acetonitrile (34851 MSDS, J.T. Baker, North America) was used to activate the desalination column, and 0.1% formic acid was used to balance the column. The sample was loaded onto the column, and then 0.1% formic acid was used to wash the column. Impurities were washed off, and finally 70% acetonitrile was used to elute. The flow-through was collected and freeze-dried. TMT reagent from TMT10plex™ Isobaric Label Reagent Set (90111, Thermo Fisher Scientific, USA) was added to 100μg digested samples and reacted at room temperature for 1h. Ammonia (013–23355, Wako Pure Chemical Industries Ltd, Japan) was added to terminate the reaction. The labeled samples were mixed followed by vortex oscillation, centrifugation, and vacuum freeze centrifugal drying. The mixed labeled samples were dissolved with 100 μ mobile phase A followed by 14000g centrifuge for 20 minutes. Then the supernatant was taken, and high-performance liquid phase was used for classification treatment. Nanoflow LC-MS/MS analysis of tryptic peptides was conducted on a quadrupole Orbitrap mass spectrometer (Q Exactive HF-X, Thermo Fisher Scientific, Bremen, Germany) coupled to an EASY nLC 1200 ultra-high pressure system (Thermo Fisher Scientific) via a nano-electrospray ion source. The mass spectrometer was operated in "top-40" data-dependent mode. The full scan range of the mass spectrum was m/z 350–1500. The resolution of the primary mass spectrometry was set to 120000 (200 m/z), with an AGC of $4e^5$, and the maximum injection time of C-trap is 50 ms. The 'Top Speed' mode was used in the secondary mass spectrometry detection. The resolution of the secondary mass spectrometry was set to 30000 (200 m/z) with the AGC of $5e^4$ and a maximum injection time of 54 ms. The collision energy of the peptide fragment was set to 36%. The mass spectrometry detection raw data (.raw) was generated.

## Transcriptome and proteome conjoint analysis

According to the quantitative and differential screening results obtained by separate analysis of transcriptome and proteome, the data of transcriptome and proteome were matched. The correlation analysis of transcriptome and proteome identification abundance was realized by the corrgram function of R language. The correlation analysis of transcriptome and proteome ratios was calculated by the cor function of R language. In order to eliminate the influence of measurement technology and explore the potential translation regulation information, the difference between transcriptome and proteome in different samples was used to measure the difference between the fold change of transcriptome and the fold change of proteome in different samples. Functional annotation and analysis: Gene Ontology (GO), GO Slim analysis, Kyoto Encyclopedia of Genes and Genomes (KEGG) analysis.

## Statistics

For western blotting, Image J was used to obtain the grayscale value of the target protein band which was further normalized with total protein. The ggpubr package of R (version 3.5.2) was applied for statistical analysis and plotting. IPP6.0 software was used to analyze the optical density of immunohistochemical photos. Wilcoxon test was used for comparisons between the two groups. $P<0.05$ was considered as statistical significance.

# Results

## Mouse model

Four mutant mice (number #2, #4, #6 and #9) were finally generated by CRISPR/Cas-mediated genome engineering. In addition, four WT even-aged mice (L161, L162, L164, and L165) on a genetic background of the C57BL/6 strain were chosen as the control.

## TEM characteristics of cardiac tissue

The cardiac tissue and cytoplasm of mutant mouse # 2 were partially dissolved, the number of organelles increased slightly, the myofibrils were broken and disappeared, the myofilament structure was obviously sparse, the residual sarcomere was symmetrically distributed, and the structure of light and dark bands was blurred. The structure of mitochondria (M), which were locally distributed between myofibrils in a small area. The complete membrane was passable, the matrix in the membrane being uniform, and the ridge being visible. Sarcoplasmic reticulum (Spr) showed no obvious expansion. The structure of Z-line (Z) and H-band (H) was slightly blurred, broken and disappeared in many places (Fig 1A).

The muscle tissue and cytoplasm of # 4, # 6, # 9 mutant mice were dissolved in a large range, the number of organelles was general, a large number of myofibrils were broken and disappeared, the structure of myofilaments was obviously loose, the residual sarcomeres were symmetrically distributed, and the structure of light and dark bands was blurred. Sarcoplasmic reticulum (Spr) slightly expanded. The structure of Z-line (Z) and H-band (H) was slightly blurred, broken and disappeared in many places, especially mutant mice # 4, # 6 (Fig 1B–1D).

The cardiac muscle tissue and cytoplasmic density of wild-type mouse L161, L162, L164, L165 were uniform, the number of organelles was general, the structure of myofibrils was slightly blurred, the arrangement was tight, the structure of myofilaments was dense, the sarcomeres were blurred, and the structure of light and dark bands was not clear. Mitochondria (M) are located between myofibrils, with a fair structure, intact membrane, uniform matrix within the membrane, dense cristae but uneven arrangement direction. Sarcoplasmic

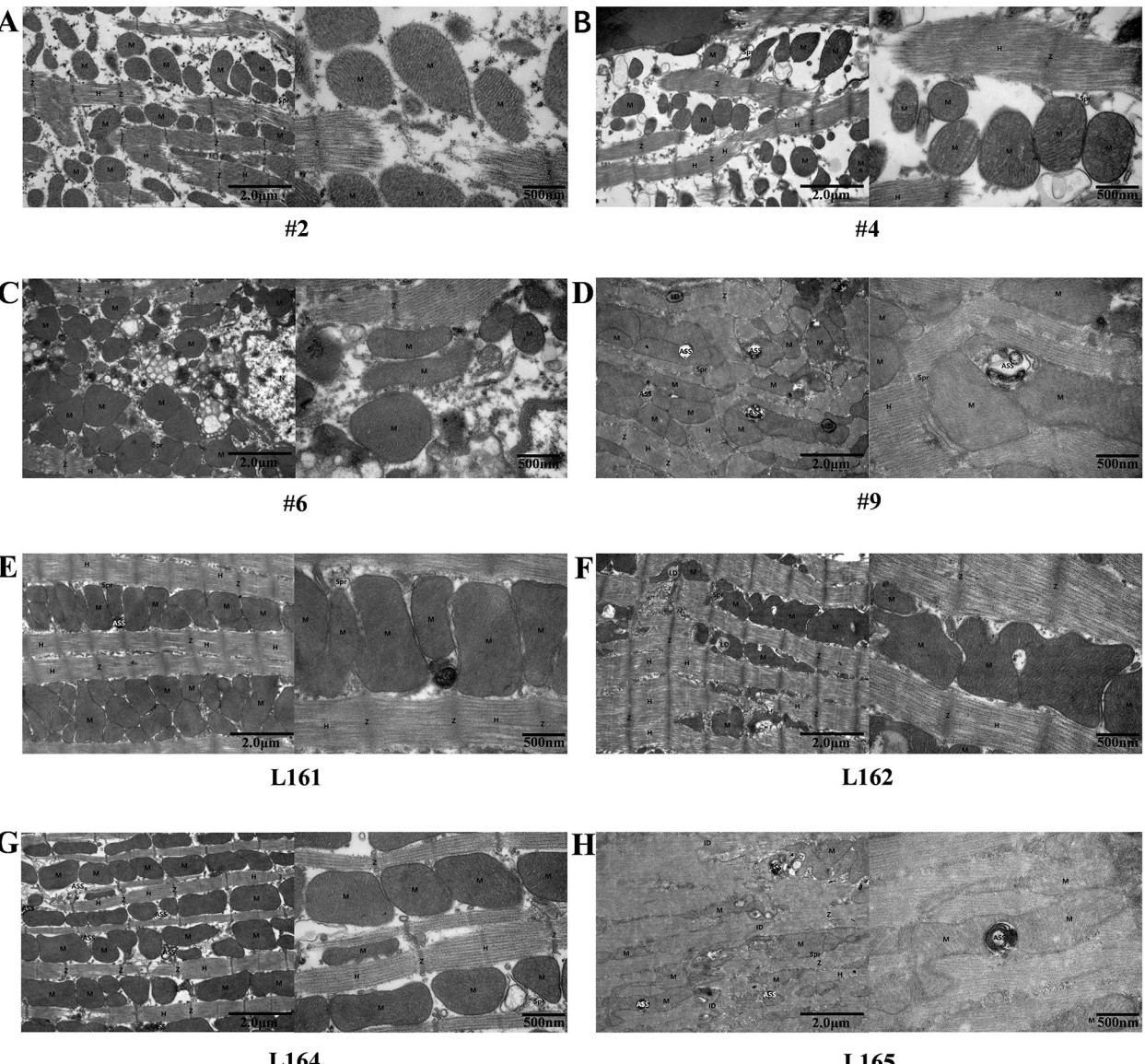

**Fig 1. Representative legends for KO and WT mice cardiac tissue observed under the TEM.** (A)~(D) Observation results of KO mice cardiac slices. The myocardial tissue of KO mice # 2, # 4, and # 6 was most severely damaged, with obvious cytoplasmic edema. More myofibrillar fibers dissolved and disappeared. A few myocardial tissues exhibited sparse myofilaments and symmetrical sarcomeres. Mitochondria were increased in number and aggregated. The myocardial damage of KO mouse # 9 was milder. (E)~(H) Observation results of WT mice cardiac slices. WT mice L161, L162, and L165 showed the least damage, with uniform cytoplasmic density and tightly arranged myofibrils. Part of the myocardial tissue had a small number of broken myofibrils and sparse muscle filaments. Most sarcomere are symmetrically distributed. The number of mitochondria was acceptable. The myocardial tissue damage of WT mouse L164 was second to that of KO mice # 2, # 4, and # 6, with slight cytoplasmic edema, uneven thickness of myofibrils, dense muscle filament structure, and symmetrical distribution of sarcomeres. The number of mitochondria had no significant increase, and their distribution was relatively uniform.

reticulum (Spr) showed no obvious expansion. The intercalated disc (ID) existed, with a longer dense area, continuous structure, and narrow intercellular space; The structure of the Z-line (Z) and H-band (H) was fuzzy and locally fractured. Autophagy lysosomes (ASS) exist (Fig 1E–1H).

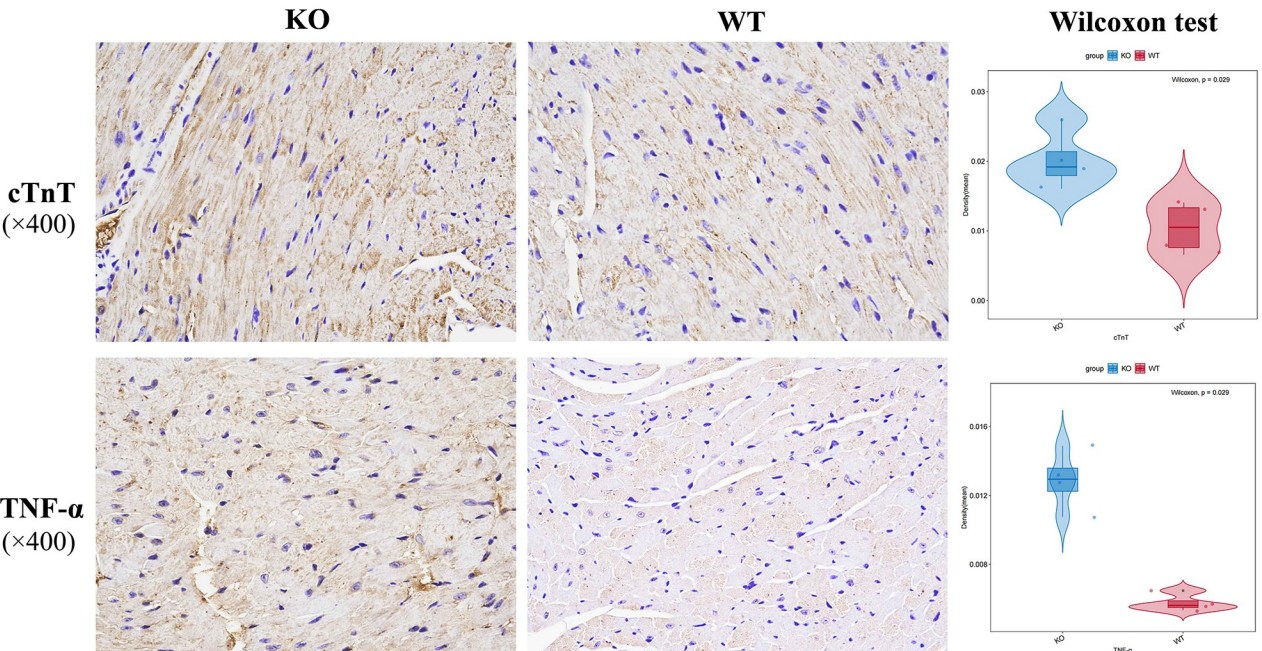

**Fig 2. Representative images of immunohistochemical staining of cTnT and TNF-α in cardiac tissue of KO and WT mice.** Blue represents the nucleus, and brownish yellow or brownish brown represents the target protein. Wilcoxon test showed that the expression level of cTnT and TNF-α was significantly upregulated in KO mice cardiac tissue than in WT mice ($P<0.05$). Scale bar, 2μm and 500 nm.

### Immunohistochemical examination of myocardial markers and inflammatory factors of cardiac tissue

The immunohistochemical staining was performed to detect the expressions of CTNT, IL6, IL10, MMP2, MMP9, TGF-β, TNF-α in myocardial tissues in WT mice and KO mice. CTNT and TNF-α were found to be significantly increased in KO mice myocardial tissues (Fig 2). In the KO mice, other markers such as IL6, IL10, MMP2, MMP9, TGF-β were have a higher trend compared to the WT mice (S1 Fig).

### Titin protein quantification of cardiac tissue

We examined the expression difference of titin isoforms (Fig 3A) between mutant mice and WT mice on protein level by WB with normalization method using total protein (Fig 3B) existing on experimental blot. Results showed no significant differences in the expression of titin proteins of 10kDa~18kDa, 18~23kDa, 23~30kDa, 30kDa, 42kDa, and 55 kDa in mutant mice heart tissue compared with WT mice heart tissue (Fig 3C–3H).

### Transcriptome analysis of two group cardiac tissue

The results of transcriptome sequencing of cardiac tissue samples from the two groups revealed significant differences in pathways associated with immune regulation and inflammatory responses (Fig 4A and 4B). Particularly, CCR6 chemokine receptor binding (GO_0031731) was significantly upregulated in the KO group by GSEA analysis (Fig 4C and 4D).

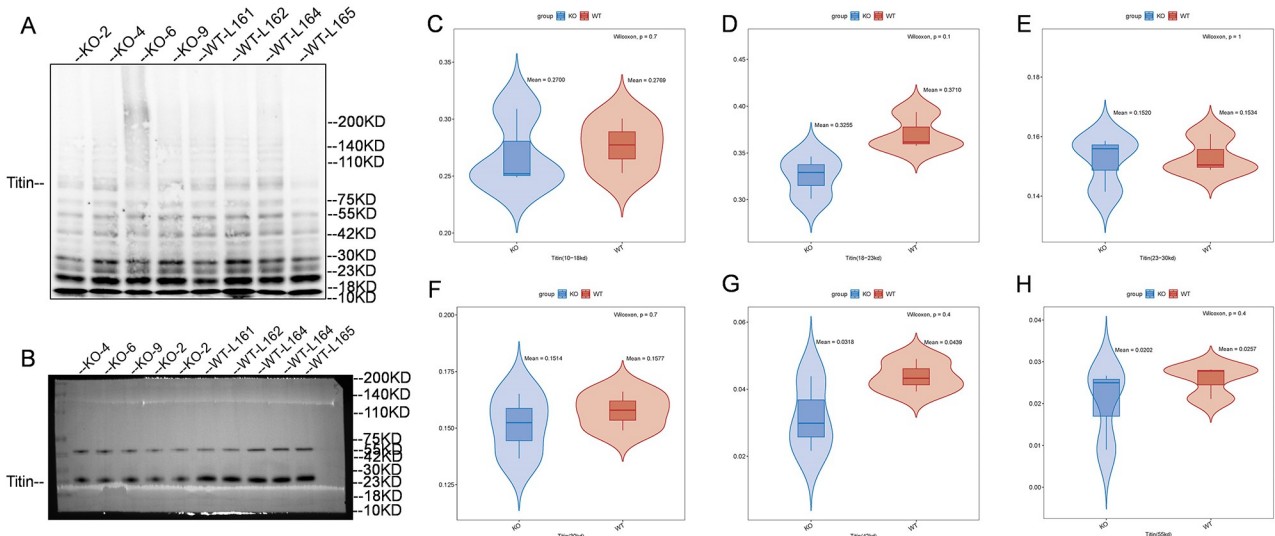

**Fig 3. Protein expression levels of Titin.** (A) Representative western blot comparing Titin levels in KO and WT mice. (B) Western blot image of total titin protein of KO mice and WT mice. (C)~(H) Wilcoxon test results of Titin of different molecular weights in KO and WT mice. There was no significant difference in the expression of Titin of 10–18 kDa, 18–23 kDa, 23–30 kDa, 30 kDa, 42 kDa and 55 kDa between KO mice and WT mice.

## TMT proteome analysis of two group cardiac tissue

PLS-DA results indicated the classification of cardiac tissue samples of mice in the two groups was better, which is well represented in both groups (Fig 5B). The protein expression of the two groups was analyzed, and the enrichment results showed that the pathways related to cardiac muscle contraction and inflammation were significantly different (Fig 5A and 5C). Heatmaps showed significant differences in protein expression between the two groups of cardiac tissue samples (Fig 5D).

## Differentially expression of transcriptome and proteome

Transcriptome and proteome data were obtained for conjoint analysis. Differentially expressed genes (DEGs) were screened and processed for GO and KEGG pathway analyses. The most enriched GO terms were classified to Biological Process (BP) and Molecular Function (MF), including muscle contraction, cardiac muscle cell development, cardiac muscle contraction, regulation of interleukin-1 production, actin filament organization, regulation of the force of heart contraction, actin binding, etc. (Fig 6A). The most enriched KEGG pathways of the DEGs were dominated by pathways involved in Dilated cardiomyopathy (DCM), vascular endothelial growth factor (VEGF), Hypertrophic cardiomyopathy (HCM), Thyroid hormone synthesis, serine/threonine kinase AMP-activated protein kinase (AMPK), HIF−1, Cardiac muscle contraction, Renin−angiotensin system, Regulation of actin cytoskeleton, etc. (Fig 6B).

## Discussion

In previous study, we found a novel *TTNtv* c.13254T>G in two probands with DCM but classified as versus uncertain significance (VUS) by American College of Medical Genetics and Genomics (ACMG). Both probands are females, aged 62 and 52 respectively. The results of echocardiography and cardiac magnetic resonance imaging both showed that proband 1 had left heart enlargement, decreased amplitude of ventricular septal and left ventricular wall

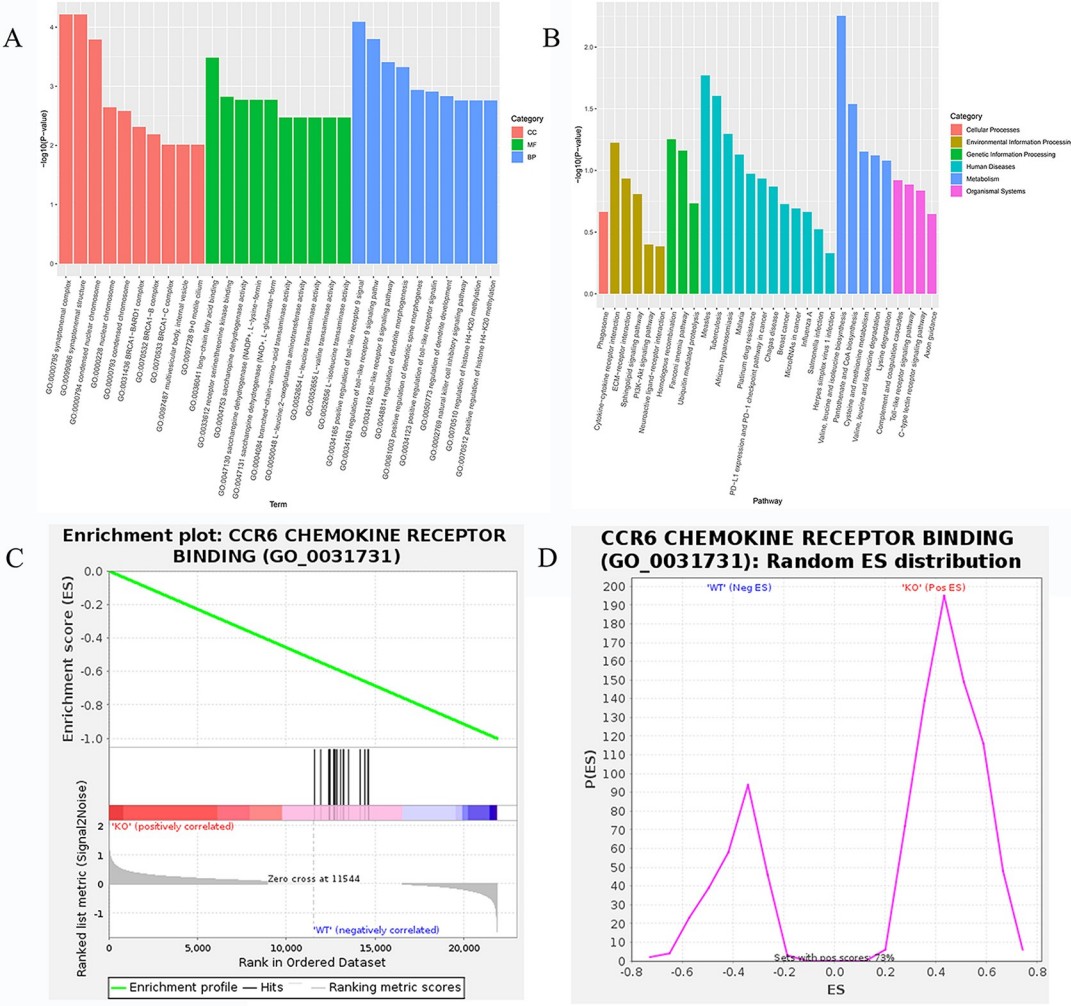

**Fig 4. GO and KEGG enrichment analyses.** (A)~(B) GO and KEGG enrichment analyses of transcriptome of two group cardiac tissue. (C)~(D) CCR6 chemokine receptor binding (GO_0031731) expression level of two group cardiac tissue by GSEA enrichment analysis of transcriptome.

movement, and decreased left ventricular systolic function. Myocardial delayed imaging showed the formation of myocardial fibrosis in the interventricular septum, left anterior wall, and inferior wall of the myocardium. The echocardiography results of proband 2 showed enlargement of the left atrium and left ventricle, and a general decrease in the amplitude of ventricular septal and left ventricular wall movement. The cardiac magnetic resonance imaging results showed left heart enlargement, segmental motion abnormalities in the left ventricular wall, severe reduction in left ventricular systolic function, and moderate reduction in right ventricular systolic function, presenting as dilated cardiomyopathy with myocardial infarction. Whole exon sequencing revealed that two probands carried a TTNtvc.13254T>G (p.Tyr4418-Ter) heterozygous mutation located in transcript NM_133379.5 (chr2:179613873). The mother of proband 1 suffered from heart disease and died of repeated heart failure. Proband 1 had 6 siblings and her sister suffered from heart disease (cardiac enlargement, details unknown). Due to the fact that proband 1 did not provide samples and clinical data of family members, it was not possible to verify whether her family members carried the mutation. The father of

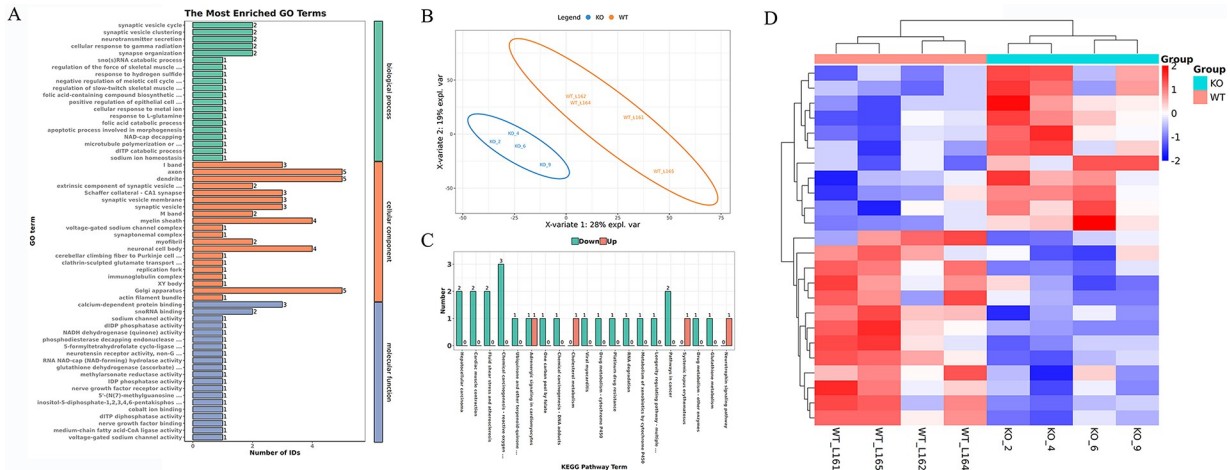

**Fig 5. TMT Proteome analysis results.** (A) The most enriched GO terms of protein in two group cardiac tissue. (B) PLS-DA analysis results of two group cardiac tissue. (C) The top 20 KEGG pathway term of the protein expression in two group cardiac tissue. (D) Protein expression heatmaps in two group cardiac tissue.

proband 2 died for unknown reasons at 40 years old, and the daughter carried the same heterozygous mutation. By searching population frequency databases including 1000 genomes, ESP 6500 and the Exome Aggregation Consortium (ExAC), *TTNtv* c.13254T>G was identified as a rare mutation. Also, the mutation could not be found in ClinVar and HGMD databases. Given the rareness and potential pathogenicity of the novel *TTNtv* c.13254T>G, we preliminarily proved that the mutation could lead to cardiac damage in a mouse model constructed by CRISPR/Cas9-mediated genome engineering through echocardiography, serological detection and histological evaluation in the former study [11]. In the current study, we made further efforts by applying multi-omics on the mouse model to supplement the pathogenic evidence of *TTNtv* c.13254T>G in cardiomyopathy.

TEM showed that the damage of cardiac tissue happened in all KO mice, with # 2, # 4 and # 6 being the most severely damaged. The cytoplasm was obviously edematous, and the myofibrils were more dissolved and disappeared. In some slices, the myofilament was sparse, the sarcomere was symmetrical, and the mitochondria increased and aggregated. However, it was observed that all WT mice had mild or almost normal myocardial damage. In these WT mice, most of the sarcomeres are symmetrically distributed, and the number of mitochondria was acceptable. These cardiac muscle TEM results and Titin protein characteristics indicated that TTN-truncating variant c.13254T>G could lead to Titin truncation expression and heart muscle damage.

The cardiac troponin I (cTnI) and T (cTnT) are established as biomarkers of cardiomyocyte injury [12]. In the diagnosis of myocardial injury, the measurement of serum cTnI and cTnT is superior to the measurement of myocardial enzymes in terms of sensitivity and specificity [13]. Immunohistochemistry results showed that cTnT was expressed much more in the cardiac issue of KO mice than in the WT mice, indicating the myocardial injury in KO mice. The TNF-α expression level of KO mice was increased much higher compared with WT mice. Additionally, in the KO mice, other markers such as IL6, IL10, MMP2, MMP9, TGF-β also had a higher trend compared to the WT mice. Inflammatory cytokines can participate in the occurrence and development of heart failure through immune and inflammatory response [14]. The RNA-seq results also indicated that CCR6 chemokine receptor binding (GO_0031731), which plays an important role in inflammatory response was significantly up-

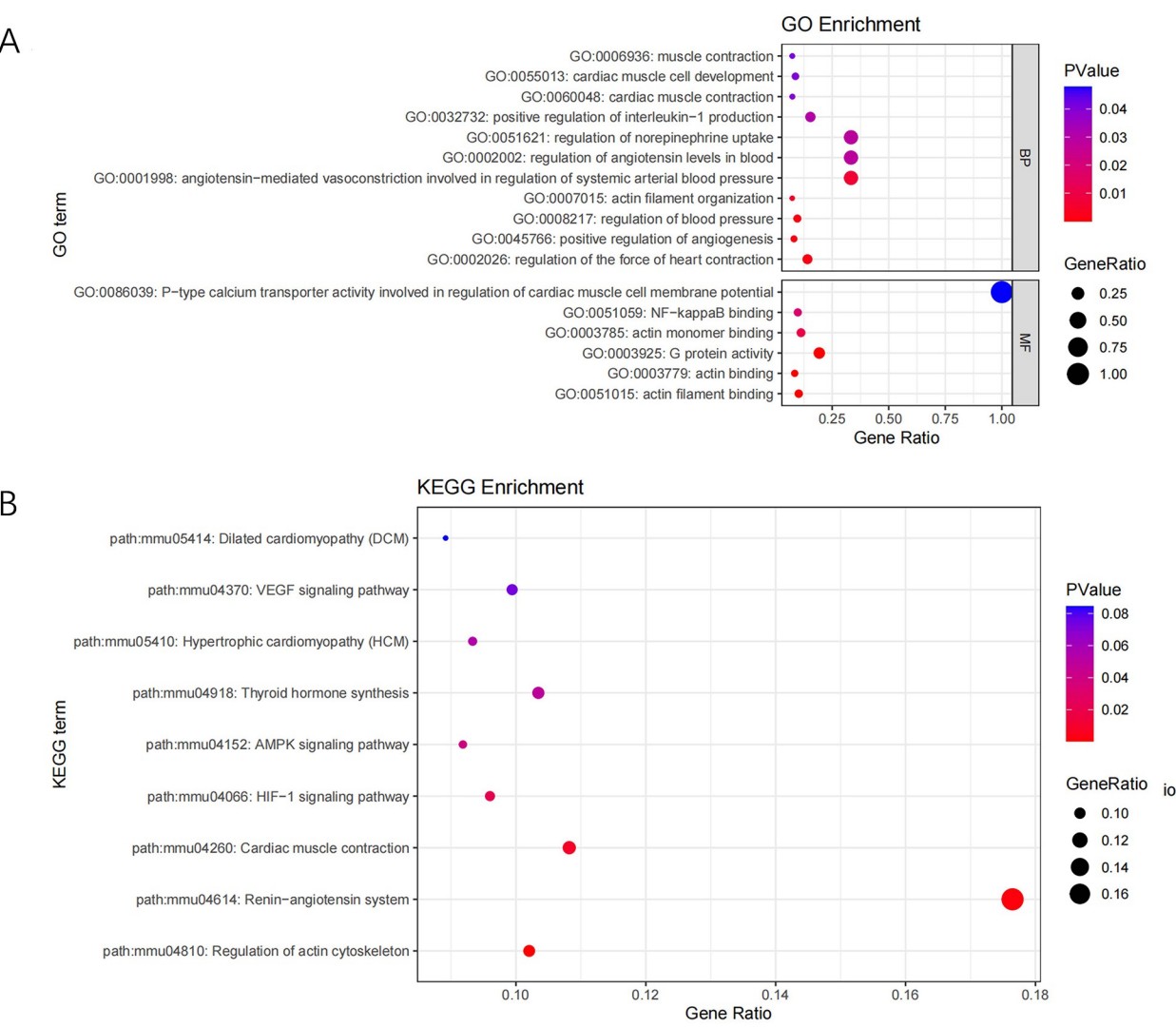

**Fig 6. GO term and KEGG pathway term enrichment of DEGs screened by conjoint analysis of RNA-seq and TMT proteome.**

regulated in the KO group by GSEA analysis in this study. Proinflammatory cytokines such as TNF-α and IL-6 stimulate cardiac oxidative stress and coronary artery dysfunction, leading to cardiac remodeling, myocardial fibrosis, and diastolic dysfunction [15]. TNF-α stimulates and induces the proliferation of cardiac fibroblasts and enhances collagen synthesis [16]. It has been reported that patients with heart failure, some of whom have dilated cardiomyopathy, have high plasma TNF-α levels [17, 18]. Accordingly, it can be inferred from the high TNF-α expression level in the cardiac tissue of KO mice that myocardial injury occurred and inflammatory response was induced.

By enrichment analysis, we found that DEGs screened from RNA-seq and proteome data mostly enriched in biological process, molecular function and pathways primarily related to development and regulation of cardiac muscle contraction, actin and actin filament binding and organization, regulation of actin cytoskeleton, DCM, HCM, manifesting that truncation of titin might affect the biological process and molecular function of these DEGs and led to myocardial injury. VEGF, AMPK pathways were also involved. Research has shown that the

expression of VEGFR in cardiac endothelial cells induces neovascularization, participates in the repair process after myocardial ischemia, prevents adverse remodeling, and reduces tissue edema and pericardial effusion [19]. And, overexpression of VEGF in cardiomyocytes can protect them from apoptosis, cause vasodilation and new vessel growth, improve local blood flow and nourish surrounding cardiomyocytes, thereby preventing tissue damage. Long term overexpression of VEGF can lead to myocardial hypertrophy and metabolic changes in cardiomyocytes [20]. Besides, AMPK is an energy sensor that controls ATP supply from substrate metabolism and protects heart from energy stress [21]. This implied that myocardial injury may activate the two pathways for heart repair and protection in the KO mice. DEGs also enriched in positive regulation of interleukin-1 (IL-1) production, which was in accordance with the immunohistochemistry results that inflammatory response occurred and TNF-α was highly expressed in the KO mice cardiac tissue. TNF is the first cytokine that appears in the blood within minutes after any injury or stress. Other pro-inflammatory mediators such as IL-1 or IL-6 appear relatively late, and there is evidence to suggest that most IL-1 or IL-6 rely on previously released TNF [22]. TNF acts through two receptors, TNFR1 and TNFR2. TNFR1 is expressed in most tissues, and the cross-linking with TNF produces a classical pro-inflammatory response, which leads to the expression of various classical pro-inflammatory cytokines, such as IL-1, IL-6, GM-CSF and so on [23]. Thus, we inferred that the increased TNF-α expression level drove the positive regulation of interleukin-1 production in KO mice.

## Conclusions

In conclusion, multi-omics methods proved that KO mice with TTNtv Y4370* had the most serious myocardial tissue injury, construct alterations, and inflammatory response. *TTN*tv c.13254T>G is a contributing factor to the myocardial injury and construct alterations in mice, and DCM in human. However, this study also has some limitations. Firstly, the number of mutant and WT mice were relatively small, which follow-up studies with large sample sizes are needed of mouse cardiac tissue and human cardiac tissue. Secondly, perhaps due to insufficient samples, not all titin protein molecule changes were observed in mutant mice. Nevertheless, we are conducting a large sample mutant mouse modeling and human cardiac tissue experiment to study the evidence of truncated expression caused by this mutation site with a deeper level.

## Supporting information

**S1 Fig. (A)~(E) The expression level of IL10, IL6, TGF-β, MMP2, MMP9 in two group cardiac tissue.**
(TIF)

**S1 Raw image. Original images for western blot.** (A) Representative gel images of different titin protein variants between KO mice and WT mice. (B) Representative gel images of total titin protein used for normalization.
(TIF)

## Acknowledgments

We are grateful for the technical support provided by BestNovo (Beijing) Medical Laboratory.

## Author Contributions

**Conceptualization:** Dan Yu, Liang Tao, Hongyan Xiao.

**Data curation:** Dan Yu, Kaisheng Lai, Hui Jiang, Zhe Liu.

**Formal analysis:** Laichun Song, Kaisheng Lai, Hui Jiang, Zhe Liu.

**Investigation:** Laichun Song.

**Project administration:** Dan Yu, Hongyan Xiao.

**Resources:** Liang Tao, Kaisheng Lai.

**Software:** Liang Tao, Kaisheng Lai.

**Supervision:** Dan Yu.

**Visualization:** Kaisheng Lai, Hongyan Xiao.

**Writing – original draft:** Hongyan Xiao.

**Writing – review & editing:** Dan Yu.

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
