## [Decision Letter · Decision Letter 0]

20 Feb 2024

PONE-D-23-40669Integrated multi-omics approach revealed TTNtv c.13254T>G causing dilated cardiomyopathy in micePLOS ONE

Dear Dr. Xiao,

Thank you for submitting your manuscript to PLOS ONE. After careful consideration, we feel that it has merit but does not fully meet PLOS ONE’s publication criteria as it currently stands. Therefore, we invite you to submit a revised version of the manuscript that addresses the points raised during the review process. While the importance of the study was recognized, significant information that is considered important is missing (phenotype of patients and mice, genetic data etc) and histological data are overrepresented without appropriate discussion. While significant changes are required for this manuscript, if you are unable to do it by the date required please request an extension.

We look forward to receiving your revised manuscript.

Kind regards,

Aldrin V. Gomes, Ph.D.

Academic Editor

PLOS ONE

 [This study is supported by Hubei Natural Science Foundation Youth Program (2022CFB841) and Scientific Research Program of Wuhan Municipal Health Commission (WX20D15). Dr Wenqing Sun received these fundings.].  

[We thank Hubei Natural Science Foundation Youth Program (2022CFB841) and Scientific Research Program of Wuhan Municipal Health Commission (WX20D15) for funding this study. We are grateful for the technical support provided by BestNovo (Beijing) Medical Laboratory.]

  [This study is supported by Hubei Natural Science Foundation Youth Program (2022CFB841) and Scientific Research Program of Wuhan Municipal Health Commission (WX20D15). Dr Wenqing Sun received these fundings.].  

6. Thank you for uploading your study's underlying data set. Unfortunately, the repository you have noted in your Data Availability statement does not qualify as an acceptable data repository according to PLOS's standards.

7. Your ethics statement should only appear in the Methods section of your manuscript. If your ethics statement is written in any section besides the Methods, please delete it from any other section. 

8. We note that Figure 1 in your submission contain copyrighted images. All PLOS content is published under the Creative Commons Attribution License (CC BY 4.0), which means that the manuscript, images, and Supporting Information files will be freely available online, and any third party is permitted to access, download, copy, distribute, and use these materials in any way, even commercially, with proper attribution. For more information, see our copyright guidelines: http://journals.plos.org/plosone/s/licenses-and-copyright.

Reviewers' comments:

Reviewer's Responses to Questions

**Comments to the Author**

1. Is the manuscript technically sound, and do the data support the conclusions?

Reviewer #1: No

Reviewer #2: Yes

2. Has the statistical analysis been performed appropriately and rigorously? 

Reviewer #1: I Don't Know

Reviewer #2: Yes

3. Have the authors made all data underlying the findings in their manuscript fully available?

Reviewer #1: No

Reviewer #2: Yes

4. Is the manuscript presented in an intelligible fashion and written in standard English?

Reviewer #1: No

Reviewer #2: Yes

5. Review Comments to the Author

Reviewer #1: Authors performed extensive experimental search to analyze functional effect but did not present and prioritize their data properly.

Many important points should be addressed to improve the quality of this manuscript.

1. There is neither clinical description of patients nor phenotype of the model mice. Authors statement “In previous study, we reported two probands diagnosed with DCM carrying a novel TTNtv c.13254T>G (p.Tyr4418Ter)…” does not supported by any data. No citation to find this previous study, no published clinical case in PubMed with this mutation under given authorships, no any minimal clinical data about patients (at least age, gender, instrumental investigation, time of follow-up, any epigenetic factors of DCM etc. No description of the phenotype in mice, time of DCM detection, heart failure progression, comparability of the human and mourine cardiac involvement.

2. Genetic data are strongly underrepresented even zygosity of the TTNtv rare variant in human and mice did not mention. The volume of genetic testing is required. Are there other candidate findings? The correct description of the variant of interest is absent (genomic coordinate with the reference genome version (hg19/hg38), correct MANE or MANE Clinical transcript number or RefSeq used for cDNA description. Considering multiple transcripts for TTN gene this information in crucial. Authors mentioned once 363-exons transcript, and it might me a virtual Meta-transcript (ENST00000589042/hg19). If so than exon 45 in Meta would not present in N2B and N2BA (principal long cardiac isoforms) and expresses only in novex-1 isoform (minor short cardiac isoform), and PSI (percentage spliced in) for this exon in myocardium as small as 1%. It raises the question about ACMG(2015) criteria chosen for this variant to get Class III of pathogenicity. How it goes in mice?

3. Does this variant undergo NMD elimination or escape NMD? This should be checked. What happens with total mRNA in human and mice? Human exon 45 (if correctly determined) in symmetric and the loss of exon 45 does not change the ORF in novex-1 transcript.

4. Detailed description of the DNA findings, RNA spectrum and quantity are important for understanding what’s going on in myocardium with this rare variant.

5. Figures with histological findings are accompanied with nice statistics plots. But it is completely unclear what parameter authors try to compare.

6. Discussion will largely depend on improved phenotypic and genetic description.

Reviewer #2: The present study revealed the association of TTNtv c. 13254T>G in dilated cardiomyopathy in murine model. The manuscript is well written in standard English and statistical analysis is properly done.

However, the following points need attention.

Major Concern.

1) Western blotting results quantification and graph (Figure 3) for different titin protein variants are not convincing and required to be re-quantified. Although the Gapdh is shown to stable between KO and WT samples, It would be more appropriated to normalize the band intensities with the total protein. Additionally, a full gel picture with molecular marker would be very helpful (either in supplement or in main manuscript).

2) Figure legends should be more informative and required to be rewritten.

Minor Concern:

1) Manuscript has some syntaxes and easily recoverable errors.

2) Method section required some more Information regarding reagents used (supplier, name of kits, catalogue no. etc).

3) At few places the selection of scientific words will be more appropriate (For example: instead of lifted ..use upregulated/ increased etc)

6. PLOS authors have the option to publish the peer review history of their article (what does this mean?). If published, this will include your full peer review and any attached files.

Reviewer #1: No

Reviewer #2: No

---

## [Author Response · Author response to Decision Letter 0]

22 Apr 2024

Dear editors and reviewers:

Thank you for your letter and for the reviewers’ comments concerning our manuscript entitle “Integrated multi-omics approach revealed TTNtv c.13254T>G causing dilated cardiomyopathy in mice” (PONE-D-23-40669). Those comments are all valuable and very helpful for revising and improving our paper, as well as the important guiding significance to our researches. We online submitted a rebuttal letter, a marked-up copy of the manuscript that highlights changes, an unmarked version of the revised paper, a list of responses (with one point by point response), a new cover letter, two reprepared figures (Fig1 and Fig3) processed by PACE, western blot figure of total Titin protein (S2 Fig), and a PDF file named “S1_raw_images” containing all original blot and gel results. We have studied comments carefully and have made correction which we hope meet with approval. Revised portion are highlighted in yellow in the paper. The main corrections in the paper and the responds to the reviewer’s comments are as following: 

Responds and rebuttal to the comments of academic editor and reviewer(s):

Journal Requirements:

1. When submitting your revision, we need you to address these additional requirements. Please ensure that your manuscript meets PLOS ONE's style requirements, including those for file naming. 

Response: Thanks for your important comments. We have examined our manuscript carefully, and the manuscript meets PLOS ONE's style requirements, including those for file naming.

Response: Thanks for your important comments. In the Methods section, we have supplemented details on (1) methods of sacrifice, (2) methods of anesthesia and/or analgesia, and (3) efforts to alleviate suffering. The content of methods of sacrifice “Mice were euthanized by cervical dislocation after anesthesia. Mice were subjected to induced anesthesia with 5% isoflurane until completely stationary, the corneal reflex disappeared, and no pain response was observed when the tail was clamped. Then, the anesthetized mice were placed on a wire mesh. The experimenter pulled the tail of the mouse with one hand, pressed the neck of the mouse with tweezers, and pulled the mouse with hands to dislocate the cervical spine” was added (Page 5, line 2-8). The content of methods of anesthesia and/or analgesia “The mice were anesthetized with a mixture of isoflurane and oxygen. After connecting the pipeline, the anesthesia induction mode with an isoflurane concentration of 3 % was set, and the diverter was opened to input the mixture of isoflurane and oxygen into the anesthesia induction box for a 1-minute induced anesthesia. After induced anesthesia, the concentration of isoflurane was adjusted to 1.5 % ~ 2 % to maintain anesthesia” was added (Page 5, line 9-15). The content of efforts to alleviate suffering “Subcutaneous injection of 0.05-0.1mg/kg buprenorphine was carried out to relieve pain when abnormal pain or urgent response occurred in mice, and intraperitoneal injection of diazepam 5 mg/kg was applied for sedation when mice were nervous” was added (Page 5, line 16-19).

Response: Thanks for your important comments. We have created a single PDF file named ‘S1_raw_images’ that contains all the original blot and gel images contained in the manuscript’s main figures. The original images have been compiled and annotated using Photoshop and then exported as a tiff file with LZW compression. The PDF file has been uploaded as a Supporting Information file. 

4. Thank you for stating the following financial disclosure: [This study is supported by Hubei Natural Science Foundation Youth Program (2022CFB841) and Scientific Research Program of Wuhan Municipal Health Commission (WX20D15). Dr Wenqing Sun received these fundings.]. Please state what role the funders took in the study. If the funders had no role, please state: ""The funders had no role in study design, data collection and analysis, decision to publish, or preparation of the manuscript."" If this statement is not correct you must amend it as needed. Please include this amended Role of Funder statement in your cover letter; we will change the online submission form on your behalf.

Response: Thanks for your important comments. Apologies for mistaking the funding. We have replaced fundings “Hubei Natural Science Foundation Youth Program (2022CFB841) and Scientific Research Program of Wuhan Municipal Health Commission (WX20D15)” by “Special Project of Hubei Provincial Health Commission (WJ2021M033)”. Dr Dan Yu received the funding. The role of the funder Dan Yu has been stated in the author contribution section of the study. Dan Yu participated in study design, conducted formal analysis and methodology. In addition, she wrote original draft and reviewed & edited the manuscript. We have provided an amended statement that declares all the funding or sources of support (whether external or internal to your organization) received during this study. We have included an amended Funding Statement within the cover letter.

5. Thank you for stating the following in the Acknowledgments Section of your manuscript: [We thank Hubei Natural Science Foundation Youth Program (2022CFB841) and Scientific Research Program of Wuhan Municipal Health Commission (WX20D15) for funding this study. We are grateful for the technical support provided by BestNovo (Beijing) Medical Laboratory.] We note that you have provided funding information that is not currently declared in your Funding Statement. However, funding information should not appear in the Acknowledgments section or other areas of your manuscript. We will only publish funding information present in the Funding Statement section of the online submission form. Please remove any funding-related text from the manuscript and let us know how you would like to update your Funding Statement. Currently, your Funding Statement reads as follows: [This study is supported by Hubei Natural Science Foundation Youth Program (2022CFB841) and Scientific Research Program of Wuhan Municipal Health Commission (WX20D15). Dr Wenqing Sun received these fundings.]. Please include your amended statements within your cover letter; we will change the online submission form on your behalf. 

Response: Thanks for your important comments. We have deleted the funding information from Acknowledgements Section of the manuscript.

6. Thank you for uploading your study's underlying data set. Unfortunately, the repository you have noted in your Data Availability statement does not qualify as an acceptable data repository according to PLOS's standards. At this time, please upload the minimal data set necessary to replicate your study's findings to a stable, public repository (such as figshare or Dryad) and provide us with the relevant URLs, DOIs, or accession numbers that may be used to access these data. For a list of recommended repositories and additional information on PLOS standards for data deposition, please see https://journals.plos.org/plosone/s/recommended-repositories.

Response: Thanks for your important comments. We have uploaded the data in the repository DRYAD database with DOI https://doi.org/10.5061/dryad.tmpg4f563 according to PLOS’s standards.

7. Your ethics statement should only appear in the Methods section of your manuscript. If your ethics statement is written in any section besides the Methods, please delete it from any other section.

Response: Thanks for your important comments. We have deleted the ethics statement written in any section besides the methods.

8. We note that Figure 1 in your submission contain copyrighted images. All PLOS content is published under the Creative Commons Attribution License (CC BY 4.0), which means that the manuscript, images, and Supporting Information files will be freely available online, and any third party is permitted to access, download, copy, distribute, and use these materials in any way, even commercially, with proper attribution. For more information, see our copyright guidelines: http://journals.plos.org/plosone/s/licenses-and-copyright. We require you to either (1) present written permission from the copyright holder to publish these figures specifically under the CC BY 4.0 license, or (2) remove the figures from your submission: a. You may seek permission from the original copyright holder of Figure 1 to publish the content specifically under the CC BY 4.0 license. We recommend that you contact the original copyright holder with the Content Permission Form (http://journals.plos.org/plosone/s/file?id=7c09/content-permission-form.pdf) and the following text:“I request permission for the open-access journal PLOS ONE to publish XXX under the Creative Commons Attribution License (CCAL) CC BY 4.0 (http://creativecommons.org/licenses/by/4.0/). Please be aware that this license allows unrestricted use and distribution, even commercially, by third parties. Please reply and provide explicit written permission to publish XXX under a CC BY license and complete the attached form.” Please upload the completed Content Permission Form or other proof of granted permissions as an ""Other"" file with your submission. In the figure caption of the copyrighted figure, please include the following text: “Reprinted from [ref] under a CC BY license, with permission from [name of publisher], original copyright [original copyright year].” b. If you are unable to obtain permission from the original copyright holder to publish these figures under the CC BY 4.0 license or if the copyright holder’s requirements are incompatible with the CC BY 4.0 license, please either i) remove the figure or ii) supply a replacement figure that complies with the CC BY 4.0 license. Please check copyright information on all replacement figures and update the figure caption with source information. If applicable, please specify in the figure caption text when a figure is similar but not identical to the original image and is therefore for illustrative purposes only.

Response: Thanks for your important comments. Although we do not know why there are copyrighted images in Figure 1, we guarantee that we own the original copyright of all images in Figure 1. This problem might be due to the unknown parameter error that occurred in the use of drawing software during the preparation of Fig 1. Now, Fig 1 has been reprepared and uploaded. We apologize for the error. 

Comments to the Author:

1. Is the manuscript technically sound, and do the data support the conclusions? The manuscript must describe a technically sound piece of scientific research with data that supports the conclusions. Experiments must have been conducted rigorously, with appropriate controls, replication, and sample sizes. The conclusions must be drawn appropriately based on the data presented. 

Reviewer #1: No

Reviewer #2: Yes

Response: Thanks for your important comments. Thank you very much.

2. Has the statistical analysis been performed appropriately and rigorously?

Reviewer #1: I Don't Know

Reviewer #2: Yes

Response: Thanks for your important comments. We have examined our statistical analysis program of the manuscript carefully, and make sure that the statistical analysis appropriately and rigorously.

3. Have the authors made all data underlying the findings in their manuscript fully available? The PLOS Data policy requires authors to make all data underlying the findings described in their manuscript fully available without restriction, with rare exception (please refer to the Data Availability Statement in the manuscript PDF file). The data should be provided as part of the manuscript or its supporting information, or deposited to a public repository. For example, in addition to summary statistics, the data points behind means, medians and variance measures should be available. If there are restrictions on publicly sharing data—e.g. participant privacy or use of data from a third party—those must be specified.

Reviewer #1: No

Reviewer #2: Yes

Response: Thanks for your important comments. We have uploaded all data as PLOS Data policy required. Thank you very much. 

4. Is the manuscript presented in an intelligible fashion and written in standard English? PLOS ONE does not copyedit accepted manuscripts, so the language in submitted articles must be clear, correct, and unambiguous. Any typographical or grammatical errors should be corrected at revision, so please note any specific errors here.

Reviewer #1: No

Reviewer #2: Yes

Response: Thanks for your important comments. We have made every effort to polish the language. Thank you very much.

5. Review Comments to the Author:

Reviewer #1

Reviewer #1: Authors performed extensive experimental search to analyze functional effect but did not present and prioritize their data properly. Many important points should be addressed to improve the quality of this manuscript.

Response: Thanks for your important comments. Thank you very much.

Comments to review：

1. There is neither clinical description of patients nor phenotype of the model mice. Authors statement “In previous study, we reported two probands diagnosed with DCM carrying a novel TTNtv c.13254T>G (p.Tyr4418Ter)…” does not supported by any data. No citation to find this previous study, no published clinical case in PubMed with this mutation under given authorships, no any minimal clinical data about patients (at least age, gender, instrumental investigation, time of follow-up, any epigenetic factors of DCM etc. No description of the phenotype in mice, time of DCM detection, heart failure progression, comparability of the human and mourine cardiac involvement.

Response: Thanks for your important comments. We are very sorry for forgetting to cite our previous study “The TTN p. Tyr4418Ter mutation causes cardiomyopathy in human and mice”. Now the previous study has been cited in the introduction section (page 3, line 24) and discussion section (Page 17, line 5). In the previous study, the clinical data about two probands was described in detail in the case reports section. The construction process of KO mice, validation by sanger sequencing, serological detection time, echocardiography detection time, histological staining time were also described in the materials and methods section. 

2. Genetic data are strongly underrepresented even zygosity of the TTNtv rare variant in human and mice did not mention. The volume of genetic testing is required. Are there other candidate findings? The correct description of the variant of interest is absent (genomic coordinate with the reference genome version (hg19/hg38), correct MANE or MANE Clinical transcript number or RefSeq used for cDNA description. Considering multiple transcripts for TTN gene this information in crucial. Authors mentioned once 363-exons transcript, and it might me a virtual Meta-transcript (ENST00000589042/hg19). If so than exon 45 in Meta would not present in N2B and N2BA (principal long cardiac isoforms) and expresses only in novex-1 isoform (minor short cardiac isoform), and PSI (percentage spliced in) for this exon in myocardium as small as 1%. It raises the question about ACMG (2015) criter

---

## [Decision Letter · Decision Letter 1]

14 Jun 2024

PONE-D-23-40669R1Integrated multi-omics approach revealed TTNtv c.13254T>G causing dilated cardiomyopathy in micePLOS ONE

Dear Dr. Xiao,

Thank you for submitting your manuscript to PLOS ONE. After careful consideration, we feel that it has merit but does not fully meet PLOS ONE’s publication criteria as it currently stands. Therefore, we invite you to submit a revised version of the manuscript that addresses the points raised during the review process.

Concerns about figure 3 and the quantification still exists, as well as the documentation of this quantification in the text is inadequate. 

We look forward to receiving your revised manuscript.

Kind regards,

Aldrin V. Gomes, Ph.D.

Academic Editor

PLOS ONE

Journal Requirements:

Reviewers' comments:

Reviewer's Responses to Questions

**Comments to the Author**

1. If the authors have adequately addressed your comments raised in a previous round of review and you feel that this manuscript is now acceptable for publication, you may indicate that here to bypass the “Comments to the Author” section, enter your conflict of interest statement in the “Confidential to Editor” section, and submit your "Accept" recommendation.

Reviewer #2: (No Response)

2. Is the manuscript technically sound, and do the data support the conclusions?

Reviewer #2: Partly

3. Has the statistical analysis been performed appropriately and rigorously? 

Reviewer #2: Yes

4. Have the authors made all data underlying the findings in their manuscript fully available?

Reviewer #2: Yes

5. Is the manuscript presented in an intelligible fashion and written in standard English?

Reviewer #2: Yes

6. Review Comments to the Author

Reviewer #2: Thank you for some of recommended changes you have made throughout the manuscript; however, the following points need to be addressed.

Major point:

1) Figure 3: Again, if you look carefully, quantification of 10-18, 18-23, 23-30, 30 KD titin proteins with GAPDH is not correct and convincible. They need to be quantify with adequate precision.

2) There is missing information regarding the software or program and method used for this quantification in text.

3) To enhance the confidence in Western results, it was suggested that to quantify each titin protein isomer with normalization method using total protein existing on experimental blot. However, it was partially done.

7. PLOS authors have the option to publish the peer review history of their article (what does this mean?). If published, this will include your full peer review and any attached files.

Reviewer #2: No

---

## [Author Response · Author response to Decision Letter 1]

26 Jun 2024

Dear editors and reviewers:

Thank you for your letter and for the reviewers’ comments concerning our manuscript entitled “Integrated multi-omics approach revealed TTNtv c.13254T>G causing dilated cardiomyopathy in mice” (PONE-D-23-40669). Those comments are all valuable and very helpful for revising and improving our paper, as well as the important guiding significance to our researches. We online submitted a rebuttal letter, a marked-up copy of the manuscript that highlights changes, an unmarked version of the revised paper, one reprepared figure (Fig3) processed by PACE. We have studied comments carefully and have made correction which we hope meet with approval. Revised portion are highlighted in yellow in the paper. The main corrections in the paper and the responds to the reviewer’s comments are as following: 

Responds and rebuttal to the comments of academic editor and reviewer(s):

Journal Requirements:

Response: Thanks for your important comments. We have reviewed the reference list carefully. It is complete and correct without any retracted papers cited. No changes have been made to the reference list.

Comments to the Author:

1. If the authors have adequately addressed your comments raised in a previous round of review and you feel that this manuscript is now acceptable for publication, you may indicate that here to bypass the “Comments to the Author” section, enter your conflict of interest statement in the “Confidential to Editor” section, and submit your "Accept" recommendation.

Reviewer #2: (No Response)

Response: Thanks for your important comments. Thank you very much.

2. Is the manuscript technically sound, and do the data support the conclusions?

Reviewer #2: Partly

Response: Thanks for your important comments. According to your suggestion, we reevaluated WB data using total protein normalization method.

3. Has the statistical analysis been performed appropriately and rigorously?

Reviewer #2: Yes

Response: Thanks for your important comments. Thank you very much. 

4. Have the authors made all data underlying the findings in their manuscript fully available?

Reviewer #2: Yes

Response: Thanks for your important comments. Thank you very much.

5. Is the manuscript presented in an intelligible fashion and written in standard English?

Reviewer #2: Yes

Response: Thanks for your important comments. Thank you very much.

6. Review Comments to the Author:

Reviewer #2: Thank you for some of recommended changes you have made throughout the manuscript; however, the following points need to be addressed.

Major point:

1) Figure 3: Again, if you look carefully, quantification of 10-18, 18-23, 23-30, 30 KD titin proteins with GAPDH is not correct and convincible. They need to be quantify with adequate precision. 

Response: Thanks for your important comments. We apologize for not understanding your suggestions correctly for the first time. We’ve re-quantified 10-18, 18-23, 23-30, 30, 42 and 55 KD titin proteins with total protein existing on experimental blot. Figure 3 has been redrawn with the updated normalized data. (Page 14, line 8-19).

2) There is missing information regarding the software or program and method used for this quantification in text.

Response: Thanks for your important comments. The information of the software and method used for WB quantification has been supplemented to the “Statistics” section. “For western blotting, Image J was used to obtain the grayscale value of the target protein band which was further normalized with total protein” was added (Page 11, line 13-14).

3) To enhance the confidence in Western results, it was suggested that to quantify each titin protein isomer with normalization method using total protein existing on experimental blot. However, it was partially done.

Response: Thanks for your important comments. We apologize for not understanding your suggestions correctly for the first time. According to your advice, we’ve normalized 10-18, 18-23, 23-30, 30, 42 and 55 KD titin proteins using total protein existing on experimental blot. 

7. PLOS authors have the option to publish the peer review history of their article (what does this mean?). If published, this will include your full peer review and any attached files.

Do you want your identity to be public for this peer review? For information about this choice, including consent withdrawal, please see our Privacy Policy.

Reviewer #2: No 

Response: Thanks for your important comments. 

Special thanks to you for your good comments.

We tried our best to improve the manuscript and made some changes in the manuscript. According to the comments of reviewers, we have revised the manuscript and list specific changes here point by point. The changes have been highlighted in yellow in revised paper. The table line number of specific changes is based on the new revised manuscript.

We appreciate for Editors/Reviewers’ warm work earnestly, and hope that the correction will meet with approval. 

Once again, thank you very much for your comments and suggestions. 

The list of specific changes

1. Page 2, line 16

“small molecular weight” was removed.

2. Page 7, line 12

“and GAPDH was the internal reference” was removed.

3. Page 8, line 2

“; GAPDH, AB-P-R 001, GOODHERE BIOTECH, Hangzhou, China” was removed.

4. Page 11, line 13-14

“For western blotting, Image J was used to obtain the grayscale value of the target protein band which was further normalized with total protein.” was added.

5. Page 14, line 10-11

“GAPDH as the internal reference” was replaced by “normalization method using total protein existing on experimental blot”.

6. Page 14, line 11-13

“Results suggested that the expression of titin protein of 10kDa~18kDa, 18~23kDa, 23~30kDa, 30kDa, 42kDa, and 55 kDa significantly decreased in mutant mice heart tissue than in WT mice heart tissue (Figs 3B-3G)” was replaced by “Results showed a downward trend in the expression of titin proteins of 10kDa~18kDa, 18~23kDa, 23~30kDa, 30kDa, 42kDa, and 55 kDa in mutant mice heart tissue compared with WT mice heart tissue (Figs 3B-3G)”.

7. Page 14, line 13

“Additionally, the total TTN protein of KO mice was significantly lower than that of WT mice (S2 Fig).” was removed.

8. Page 14, line 16-19

“The expression of Titin of 10-18 kDa, 18-23 kDa, 23-30 kDa, 30 kDa exhibited an apparent decline in cardiac tissue of KO mice compared with WT mice. 42 kDa and 55 kDa titin of KO mice expressed a decreasing trend” was replaced by “There was no significant difference in the expression of Titin of 10-18 kDa, 18-23 kDa, 23-30 kDa, 30 kDa, 42 kDa and 55 kDa between KO mice and WT mice. But all these 6 titin isoforms expressed a downward trend in heart tissue of KO mice.”.

9. Page 17, line 23-24

“western blot results suggested a significant decrease in the protein expression level of some Titin isoforms caused by TTN-truncating variant c.13254T>G, especially low molecular weight Titin protein fragments” was replaced by “western blot results suggested a downward trend in the protein expression level of some Titin isoforms caused by TTN-truncating variant c.13254T>G”.

10. Page 20, line 3

“small molecular” was removed.

11. Page 23, line 11-12

“S2 Fig. (A) Quantification results of the total titin of KO mice and WT mice by WB. WB was carried out twice on KO mouse #2 and WT mouse L164 heart tissue samples. (B) Wilcoxon test showed that titin's total protein expression level was profoundly declined in KO mice than in WT mice.” was removed.

---

## [Decision Letter · Decision Letter 2]

14 Aug 2024

PONE-D-23-40669R2Integrated multi-omics approach revealed TTNtv c.13254T>G causing dilated cardiomyopathy in micePLOS ONE

Dear Dr. Xiao,

Thank you for submitting your manuscript to PLOS ONE. After careful consideration, we feel that it has merit but does not fully meet PLOS ONE’s publication criteria as it currently stands. Therefore, we invite you to submit a revised version of the manuscript that addresses the points raised during the review process.

Thank you for the revised version of your manuscript. Please address two minor comments by one of the reviewers.

We look forward to receiving your revised manuscript.

Kind regards,

Aldrin V. Gomes, Ph.D.

Academic Editor

PLOS ONE

Journal Requirements:

Reviewers' comments:

Reviewer's Responses to Questions

**Comments to the Author**

1. If the authors have adequately addressed your comments raised in a previous round of review and you feel that this manuscript is now acceptable for publication, you may indicate that here to bypass the “Comments to the Author” section, enter your conflict of interest statement in the “Confidential to Editor” section, and submit your "Accept" recommendation.

Reviewer #2: (No Response)

Reviewer #3: All comments have been addressed

2. Is the manuscript technically sound, and do the data support the conclusions?

Reviewer #2: Partly

Reviewer #3: Yes

3. Has the statistical analysis been performed appropriately and rigorously? 

Reviewer #2: Yes

Reviewer #3: Yes

4. Have the authors made all data underlying the findings in their manuscript fully available?

Reviewer #2: Yes

Reviewer #3: Yes

5. Is the manuscript presented in an intelligible fashion and written in standard English?

Reviewer #2: Yes

Reviewer #3: Yes

6. Review Comments to the Author

Reviewer #2: Thanks for suggested changes: However, there are minor concern and need to be addressed.

(1) It would be better to submit complete gel picture of total protein (used for normalization) either under the main Western Gel in Figure or in Supplement.

(2) Statistically, there is no differences in titin protein isomers between KO and WT. Moreover, there is negligible differences in the mean values of quantifications of all titin isomers between KO and WT. In that case, it not appropriate to describe them as "decreasing trend" in the text and figure legend. Critically, there is no changes. It is recommended to make changes and describe accordingly in results, discussion, and figure legend.

Reviewer #3: The author has adequately addressed the comments of the reviewers from the previous revision. The statistics have been performed appropriately and rigorously, and the manuscript is presented in an intelligible fashion and written in standard English.

7. PLOS authors have the option to publish the peer review history of their article (what does this mean?). If published, this will include your full peer review and any attached files.

Reviewer #2: No

Reviewer #3: No

---

## [Author Response · Author response to Decision Letter 2]

29 Aug 2024

Dear editors and reviewers:

Thank you for your letter and for the reviewers’ comments concerning our manuscript entitled “Integrated multi-omics approach revealed TTNtv c.13254T>G causing dilated cardiomyopathy in mice” (PONE-D-23-40669). Those comments are all valuable and very helpful for revising and improving our paper, as well as the important guiding significance to our researches. We online submitted a rebuttal letter, a marked-up copy of the manuscript that highlights changes, an unmarked version of the revised paper, one re-prepared figure (Figure 3) processed by PACE, and an updated S1_raw_images. We have studied comments carefully and have made correction which we hope meet with approval. Revised portion are highlighted in yellow in the paper. The main corrections in the paper and the responds to the reviewer’s comments are as following: 

Responds and rebuttal to the comments of academic editor and reviewer(s):

Journal Requirements:

Response: Thanks for your important comments. We have reviewed the reference list carefully. It is complete and correct without any retracted papers cited. No changes have been made to the reference list.

Comments to the Author:

1. If the authors have adequately addressed your comments raised in a previous round of review and you feel that this manuscript is now acceptable for publication, you may indicate that here to bypass the “Comments to the Author” section, enter your conflict of interest statement in the “Confidential to Editor” section, and submit your "Accept" recommendation.

Reviewer #2: (No Response)

Reviewer #3: All comments have been addressed

Response: Thanks for your important comments. Thank you very much.

2. Is the manuscript technically sound, and do the data support the conclusions?

Reviewer #2: Partly

Reviewer #3: Yes

Response: Thanks for your important comments. According to your suggestion, we’ve supplemented the gel image of the total protein used for normalization and deleted WT mice titin isoforms decreasing trend-related content from the article.

3. Has the statistical analysis been performed appropriately and rigorously?

Reviewer #2: Yes

Reviewer #3: Yes

Response: Thanks for your important comments. Thank you very much. 

4. Have the authors made all data underlying the findings in their manuscript fully available?

Reviewer #2: Yes

Reviewer #3: Yes

Response: Thanks for your important comments. Thank you very much.

5. Is the manuscript presented in an intelligible fashion and written in standard English?

Reviewer #2: Yes

Reviewer #3: Yes

Response: Thanks for your important comments. Thank you very much.

6. Review Comments to the Author:

Reviewer #2: Thanks for suggested changes: However, there are minor concern and need to be addressed.

Major point:

(1) It would be better to submit complete gel picture of total protein (used for normalization) either under the main Western Gel in Figure or in Supplement. 

Response: Thanks for your important comments. The gel picture of the total protein used for normalization has been added to figure 3 as figure 3B, and to the supplementary file S1_raw_images. We first obtained gray values of titin isoforms and total protein using the software ImageJ. Then, titin isoforms were normalized by being divided by the gray value of the total protein. We further analyzed the difference in the normalized gray value of titin isoforms between KO mice and WT mice. The results showed no significant difference in the protein expression level of these titin isoforms between the two groups. 

(2) Statistically, there is no differences in titin protein isomers between KO and WT. Moreover, there is negligible differences in the mean values of quantifications of all titin isomers between KO and WT. In that case, it not appropriate to describe them as "decreasing trend" in the text and figure legend. Critically, there is no changes. It is recommended to make changes and describe accordingly in results, discussion, and figure legend.

Response: Thanks for your important comments. We’ve removed all the conclusions related to the decreasing trend of WT mice titin isoforms from the abstract, results, discussion and figure legend of the manuscript (Page 2, line 16; Page 14, line 10-13; Page 14, line 19; Page 17, line 22; Page 20, line 2).

Reviewer #3: The author has adequately addressed the comments of the reviewers from the previous revision. The statistics have been performed appropriately and rigorously, and the manuscript is presented in an intelligible fashion and written in standard English. 

Response: Thanks for your important comments. 

7. PLOS authors have the option to publish the peer review history of their article (what does this mean?). If published, this will include your full peer review and any attached files.

Do you want your identity to be public for this peer review? For information about this choice, including consent withdrawal, please see our Privacy Policy.

Reviewer #2: No

Reviewer #3: No

Response: Thanks for your important comments. 

Special thanks to you for your good comments.

We tried our best to improve the manuscript and made some changes in the manuscript. According to the comments of reviewers, we have revised the manuscript and list specific changes here point by point. The changes have been highlighted in yellow in revised paper. The table line number of specific changes is based on the new revised manuscript.

We appreciate for Editors/Reviewers’ warm work earnestly, and hope that the correction will meet with approval. 

Once again, thank you very much for your comments and suggestions. 

The list of specific changes

1. Page 2, line 16

The sentence “The TTNtv Y4370* caused a decrease in some isoforms of Titin protein.” was removed.

2. Page 14, line 8

“isoforms (Fig 3A)” was added.

3. Page 14, line 10

“(Fig 3B)” was added. “(Figs 3A)” was deleted.

4. Page 14, line 10-13

Sentence “Results showed a downward trend in the expression of titin proteins of 10kDa~18kDa, 18~23kDa, 23~30kDa, 30kDa, 42kDa, and 55 kDa in mutant mice heart tissue compared with WT mice heart tissue (Figs 3B-3G)” was replaced by “Results showed no significant differences in the expression of titin proteins of 10kDa~18kDa, 18~23kDa, 23~30kDa, 30kDa, 42kDa, and 55 kDa in mutant mice heart tissue compared with WT mice heart tissue (Figs 3C-3H)”.

5. Page 14, line 15-16

“(B) Western blot image of total titin protein of KO mice and WT mice.” was added.

6. Page 14, line 16

“(B)~(G)” was replaced by “(C)~(H)”.

7. Page 14, line 19

“But all these 6 titin isoforms expressed a downward trend in heart tissue of KO mice.” was removed.

8. Page 17, line 22

“In addition, western blot results suggested a downward trend in the protein expression level of some Titin isoforms caused by TTN-truncating variant c.13254T>G.” was deleted.

9. Page 20, line 1

“and” was added.

10. Page 20, line 2

“, and decreased expression level of titin isoforms” was removed.

---

## [Decision Letter · Decision Letter 3]

24 Sep 2024

Integrated multi-omics approach revealed TTNtv c.13254T>G causing dilated cardiomyopathy in mice

PONE-D-23-40669R3

Dear Dr. Xiao,

We’re pleased to inform you that your manuscript has been judged scientifically suitable for publication and will be formally accepted for publication once it meets all outstanding technical requirements.

Kind regards,

Aldrin V. Gomes, Ph.D.

Academic Editor

PLOS ONE

Additional Editor Comments (optional):

Reviewers' comments:

Reviewer's Responses to Questions

**Comments to the Author**

1. If the authors have adequately addressed your comments raised in a previous round of review and you feel that this manuscript is now acceptable for publication, you may indicate that here to bypass the “Comments to the Author” section, enter your conflict of interest statement in the “Confidential to Editor” section, and submit your "Accept" recommendation.

Reviewer #2: All comments have been addressed

2. Is the manuscript technically sound, and do the data support the conclusions?

Reviewer #2: Yes

3. Has the statistical analysis been performed appropriately and rigorously? 

Reviewer #2: Yes

4. Have the authors made all data underlying the findings in their manuscript fully available?

Reviewer #2: Yes

5. Is the manuscript presented in an intelligible fashion and written in standard English?

Reviewer #2: Yes

6. Review Comments to the Author

Reviewer #2: (No Response)

7. PLOS authors have the option to publish the peer review history of their article (what does this mean?). If published, this will include your full peer review and any attached files.

Reviewer #2: No

---

## [Editor Report · Acceptance letter]

26 Sep 2024

PONE-D-23-40669R3 

PLOS ONE

Dear Dr. Xiao, 

I'm pleased to inform you that your manuscript has been deemed suitable for publication in PLOS ONE. Congratulations! Your manuscript is now being handed over to our production team.

Kind regards, 

on behalf of

Dr. Aldrin V. Gomes 

Academic Editor

PLOS ONE